# Using vision transformer-based GANs against Vision Transformers

**Pămînt Andrei-Florin**
Babeș-Bolyai University
pamintandreifl1@yahoo.com

**Sergiu Adrian Darabant**
Department of Computer Science, Babeș-Bolyai University
sergiu.darabant@ubbcluj.ro

## Abstract

Vision transformers have become one of the best architectures for image classification tasks. In this paper, we introduce a novel method for creating adversarial attacks in a black box environment without using surrogate models. Specifically, we introduce a single encoder and a three-encoder Transformer based GAN that creates a perturbation with a successful attack rate higher than state of the art methods.

## 1 Introduction

Vision Transformers (Dosovitskiy et al., 2020) have shown impressive results on the image classification tasks. In this paper we investigate their resistance to adversarial attacks. Our proposed method is a full black-box model, meaning we have zero information about the target model and we are not using any surrogate models to gather the gradients from. The only information we extract from the target model is the result vector.

## 2 State of the art

Vision transformers are a recent architectural innovation based on the state of the art solutions in the natural language processing. They transform images into sequences of patches that are represented as tokens. Multi-headed attention is applied to the tokens, providing the models with a bigger field of view over the image (Vaswani et al., 2017). Multiple architecture and training optimizations have been developed to improve the performance of the original ViT (Dosovitskiy et al., 2020) implementation notably Touvron et al. (2020) Chen et al. (2021) Zhai et al. (2021) and many more.

Adversarial attacks fall into two big categories: *white-box* and *black-box* attacks, based on the amount of information available. Most of the research on attacks against ViTs has focused on a subset of black-box attacks, the surrogate model transferable attack Wei et al. (2021) Wang et al. (2022) Naseer et al. (2021). In this type of attack, the attacker has full access to models that have a similar architecture. Generating adversarial images using similar techniques to GANs (Goodfellow et al., 2014) has been explored in Mao et al. (2020) and Baluja & Fischer (2017).

Transformer-based GANs haven't seen much interest thus the only notable mention is the paper that popularized them, Lee et al. (2021).

## 3 Three Encoder Adversarial Transformer GAN

Our method consists of a Transformer based GAN that generates the perturbation based on two images $x_{attack}$ and $x_{defense}$. $x_{defense}$ represents the image we are attacking, $x_{attack}$ is a helper image of class $y_{attack} \in Y_{attack}$ with $Y_{attack}$ being a subset of all the classes in the dataset. These images get transformed into patches, linearly projected and positional embedding are added. This mimics the implementation of the original ViT. We then pass the resulting patches into 3 standard transformer encoders as follows: The first encoder takes as input the patches from both images $x_{attack}, x_{defense}$, the second encoder gets the patches only from the $x_{attack}$ image and the third

Table 1: ASR(%). The white box surrogate model was DeiT-B when the targeted model was ViT-B and Visformer, and ViT-B was the surrogate model when DeiT-B was the targeted model

| METHOD | ViT-B | Visformer-S | DeiT-B |
|---|---|---|---|
| ATA (Wang et al., 2022) | 18.40 | 14.92 | 17.82 |
| PNA (Wei et al., 2021) | 28.67 | 28.96 | 39.80 |
| A-ViTGAN(ours) | **44.69** | **59.09** | **54.30** |
| A-ViTGAN(ours) with no attack image | *41.89* | *56.45* | *50.24* |

Table 2: Comparison of the same model with different subset sizes of classes used for the attack images in training, for the same amount of epochs. The target model was Visformer.

| Subset Size(% of total) | ASR |
|---|---|
| 5% | 43.13 |
| 10% | 44 |
| 30% | 53.08 |
| 50% | 59.09 |
| 100% | 57.29 |

encoder gets the patches only from the $x_{defense}$ image. With this three-encoder architecture, we aim to provide the model not only with information about the targeted image, but also about other images and classes and relationships between them. The result from these three encoders is concatenated and passed to a CNN similar to the generator introduced in Radford et al. (2015) but with convolution layers weaved between the transpose layers. This is done to upscale to the original image size of 224x224x3. The resulting matrix from the CNN is our perturbation. To limit the magnitude of changes to the original image we are normalizing the perturbation using the formula from Mao et al. (2020). $\Delta x = \Delta x \min(1, \frac{\epsilon}{\|\Delta x\|_p})$ where $\Delta x$ is the perturbation, and $\epsilon$ is the maximum $L_p$ norm (threshold) we are setting for the attack. This is a hyper-parameter of the system. Based on the findings of Sharif et al. (2018) which show that all three, $L_0, L_2$ and $L_\infty$ have their shortcomings but $L_2$ has the least potential to impact humans in correctly classifying the image, we chose to normalize using $L_2$.

## 4 EXPERIMENTAL RESULTS

All experiments were conducted on Imagenet, this is because it better resembles real-life attack scenarios than other datasets. We will use ASR(attack success rate) (Wei et al., 2021) as the metric to compare the effectiveness of different models. We set our $\epsilon = 20$ and use a patch size $P = 16$. Table 5 shows the effectiveness of our model compared to the current best-performing attacks on ViTs. The results from ATA (Wang et al., 2022) and PNA (Wei et al., 2021) are lower than stated in their respective papers because we set a more restrictive norm and because we chose the surrogate model without trying to improve the results. We didn't try to optimize their attacks by choosing good surrogate models. The attacked model is *unknown* in this case, hence it is not possible to choose similar surrogate models. A-ViTGAN with no attack image represents a basic version of our method which only takes as input the targeted image and has only one encoder. In a real attack scenario, the attacker might not have information on all classes used for training. Table 2 shows the effects of the size of $Y_{attack}$ on performance. The results show that even with as little as 5% of the classes the model still performs better than previous methods but for the three encoder strategies one needs at least 50% of classes available to have a performance boost over the no-attack image method.

## 5 CONCLUSION

In conclusion, we have shown that Transformer based GANs outperform current attack methods against ViTs and introduced a novel three-encoder Transformer GAN that supports our proposal and shows great promise.

URM STATEMENT

Pămînt Andrei-Florin meets the URM criteria of ICLR 2023 Tiny Papers Track.

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

# A APPENDIX

Github link: `https://github.com/pamintandrei/A-ViTGAN`

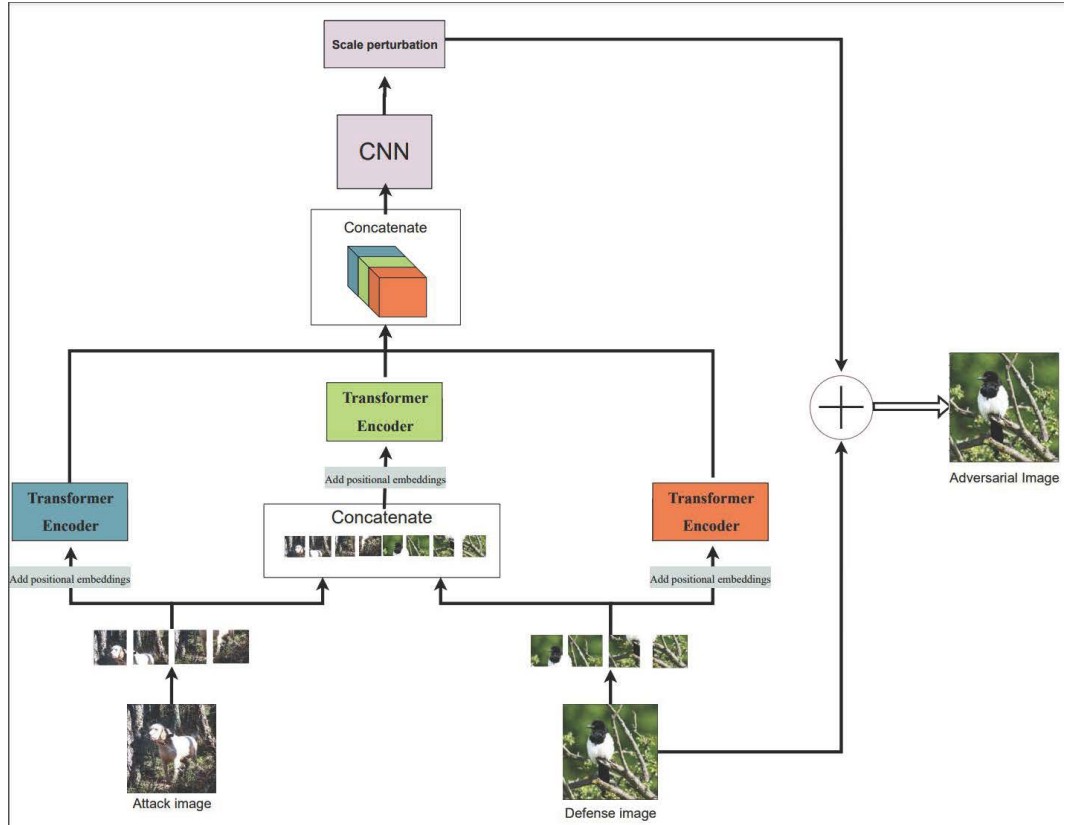

Figure 1: Architecture overview

Table 3: Results of training with the same subset size- five different randomly sampled 50% subsets out of the original dataset. Shows that the attacked classes have no significant impact on the results.

| Subset indices | ASR |
|---|---|
| 1 | 58.79 |
| 2 | 58.65 |
| 3 | 59.09 |
| 4 | 58.62 |
| 5 | 58.82 |

Table 4: ASR(%) for targeted attacks, the loss function is changed from the difference to ground truth $y$ to the difference against $y_{attack}$.

| METHOD | ViT-B | Visformer | DeiT |
|---|---|---|---|
| A-ViTGAN | 14.33 | 8.72 | 16.24 |

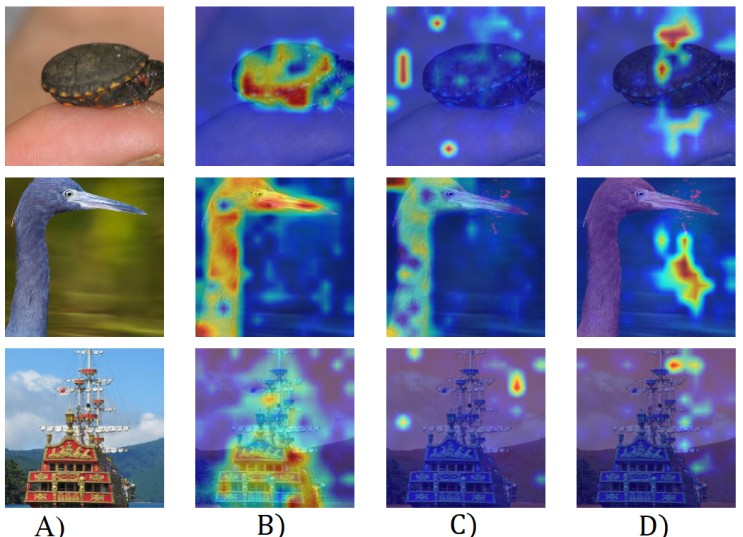

A)                    B)                    C)                    D)

Figure 2: Image A) represents the original image, B) represents the attention heatmap of the targeted model on the original, unperturbed image for the correct class C) represents the attention heatmap of the targeted model on the perturbed image, on the correct class D) represents the attention heatmap of the targeted model on the perturbed image, on the class that now has the highest confidence. The heatmaps were generated using Transformer-Explainability that is based on Chefer et al. (2020)

Table 5: ASR(%). The white box surrogate model was DeiT-B when the targeted model was ViT-B and Visformer, and ViT-B was the surrogate model when DeiT-B was the targeted model

| METHOD | ViT-B | Visformer-S | DeiT-B |
|---|---|---|---|
| ATA (Wang et al., 2022) | 18.40 | 14.92 | 17.82 |
| PNA (Wei et al., 2021) | 28.67 | 28.96 | 39.80 |
| ATN (Baluja & Fischer, 2017) | 0.9 | 0.85 | 1.42 |
| A-ViTGAN(ours) | **44.69** | **59.09** | **54.30** |
| A-ViTGAN(ours) with no attack image | *41.89* | *56.45* | *50.24* |

Table 6: Direct comparison of our method to the other attacks using their Lp norm. The attacked model is DeiT-B.

| Paper | Lp norm used | Bounding $\epsilon$ | Their results | Our results |
|---|---|---|---|---|
| PNA (Wei et al., 2021) | $L_\infty$ | 16 | 63.85 | **94.32** |
| ATA (Wang et al., 2022) | Pixel-wise $L_0$ | 1024 | 42.4 | **48.72** |

Table 7: ASR(%) with different transformer encoders enabled

| Attack Encoder | Defense Encoder | Attack+Defense encoder | ASR |
|---|---|---|---|
| ✓ | ✗ | ✗ | 40.72 |
| ✗ | ✓ | ✗ | 50.24 |
| ✗ | ✗ | ✓ | 51.62 |
| ✓ | ✓ | ✗ | 52.77 |
| ✓ | ✗ | ✓ | 53.9 |
| ✗ | ✓ | ✓ | 50.23 |
| ✓ | ✓ | ✓ | 54.3 |

Table 8: Direct comparison of our method to the other attacks using their Lp norm. The attacked model is DeiT-B.

| Paper | Lp norm used | Bounding $\epsilon$ | Their results | Our results |
|---|---|---|---|---|
| PNA (Wei et al., 2021) | $L_\infty$ | 16 | 63.85 | **94.32** |
| ATA (Wang et al., 2022) | Pixel-wise $L_0$ | 1024 | 42.4 | **48.72** |

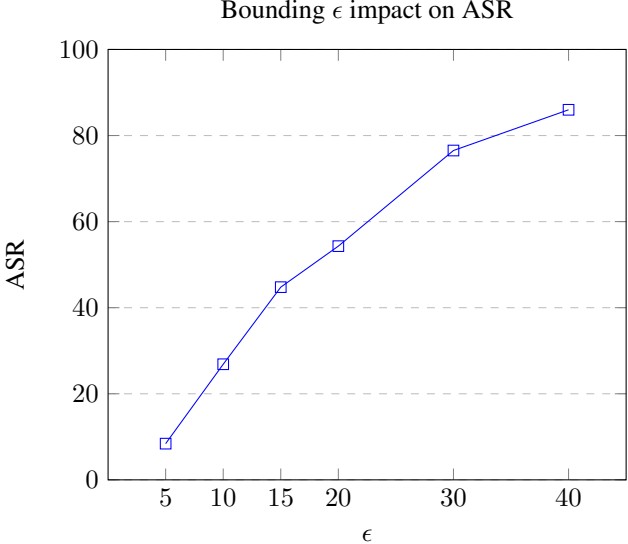

Figure 3: Impact of the bounding $\epsilon$ value on the ASR. Experimental results obtained by varying the bounding value.

