# OpenReview forum: "Using vision transformer-based GANs against Vision Transformers"
_ICLR.cc/2023/TinyPapers — Submitted to Tiny Papers @ ICLR 2023_

### Official Review · Reviewer_83ac · 2023-03-28

**Confidence:** 4

**Summary Of Contributions:**

This paper uses transformer-based GANs to generate adversarial perturbations to attack vision transformers. Results show that the proposed attack is more effective than existing black-box attacks.

**Rating:**

Great Start (GS): a submission which meets some of the reviewing criteria but has room for improvement

**Strengths And Weaknesses:**

Strengths:
- The paper presents an attack that is shown to be more effective on attacking vision transformers in a black-box setting.

Weaknesses:
- It is unclear how existing GAN-based attacks (even if they are not transformer-based) perform on the settings studied in this paper.
- It is unclear where the improvement comes from, without sufficient ablation studies.



**Suggested Changes:**

Add more experiements, as suggested in "Strengths And Weaknesses".

---

> ### Author Response · Authors · 2023-05-10
> **Response to reviewer 83ac**
>
> Thank you for taking the time to review our paper.
>
> >It is unclear how existing GAN-based attacks (even if they are not transformer-based) perform on the settings studied in this paper.
>
> We have added experimental results in  table 5 with results for ATN
>
> >It is unclear where the improvement comes from, without sufficient ablation studies.
>
> We have added an ablation study of our architecture
>
> Thank you.

---

### Official Review · Reviewer_C3kR · 2023-03-29

**Confidence:** 4

**Summary Of Contributions:**

This work introduces a new way of investigating vision transformers's resistance to adversarial attacks. The work utlizes three transformers encoders for its adversarial training and claim to outperforms all current state-of-the-art resuts.

**Rating:**

Great Start (GS): a submission which meets some of the reviewing criteria but has room for improvement

**Strengths And Weaknesses:**

Strengths
- The authors showcase an excellent technical depth by utilizing a 3-encoder architecture to ensure the model has information on not just the targeted image but also about other images and the classes and relationships between them.
- The authors also leverage nicely proven, standard techniques while acknowledging those works. The submission page limit could have caused an oversight, but that wasn't the case.

Weaknesses
- Works on which the methodology was based were cited generously. Still, I find it a bit worrying that the authors mention that there has been very little interest in Transformer-based GANs, but a quick search on Semantic Scholar proved otherwise. The work could use a more extensive literature review to clarify that efforts are not being duplicated needlessly.
- While the authors included much information about their methodology, reproducing the work might not be straightforward due to several missing details. For example, the authors mentioned using a more restrictive norm than usual, but there was no explanation of how and why a more restrictive norm was selected.

**Suggested Changes:**

While the Attack Success Rate (ASR) scores of this work seem impressive compared to ATA's and PNA's scores, the communicated results allow for skepticism on whether the hyper-parameters were hand-selected to make this work superior or not. The authors claim that ATA and PNA have lower scores due to a more restrictive norm used. An extensive ablation study can put to rest doubts about the rigor of the work.

---

> ### Author Response · Authors · 2023-05-10
> **Response to reviewer C3kR**
>
> Thank you so much for the insightful review, we have listened to your feedback and have made several improvements.
>
>
> >the communicated results allow for skepticism on whether the hyper-parameters were hand-selected to make this work superior or not.
>
> We added table 6 which compares our model with the hyper-parameters from the cited papers and has shown that we still achieve better results even when applying their settings. Even in the case of ATA where our attack isn’t optimized for L0 norm, and theirs is, we still achieve better results.
>
>
> >An extensive ablation study can put to rest doubts about the rigor of the work.
>
> We have added an ablation study for the transformer encoders and a graph for how the bounding L2 norm affects ASR.
>
>
> >Still, I find it a bit worrying that the authors mention that there has been very little interest in Transformer-based GANs, but a quick search on Semantic Scholar proved otherwise. The work could use a more extensive literature review to clarify that efforts are not being duplicated needlessly.”
>
> That comment was made based on the fact that the main Transformer-based GAN paper, ViTGAN has only 88 citations at the time of writing this comment( even less when the paper was written) and we consider that notable work on Transformer-based GANs should cite it. The paper “Transformer-based Generative Adversarial Networks in Computer Vision: A Comprehensive Survey“[1] does an extensive survey of Transformer-based GANs and doesn’t mention any that are used for adversarial attacks thus unless both (we and them) missed it, there shouldn’t be any other works tackling this problem with this approach.
>
> >“While the authors included much information about their methodology, reproducing the work might not be straightforward due to several missing details. “
>
> We have added a github link with the pytorch model to ease reproducibility. The repository is publicly available.
>
>
> >“the authors mentioned using a more restrictive norm than usual, but there was no explanation of how and why a more restrictive norm was selected.”
>
> At the end of section 3 we state “This is a hyper-parameter of the system. Based on the findings of Sharif et al. (2018) which show that all three, L0, L2  and L_infinity have their shortcomings but L2 has the least potential to impact humans in correctly classifying the image, we chose to normalize using L2.” We think that the cited paper provides enough insight into why L2 is the best Lp norm to be used. The exact value for the bounding L2 norm was selected to provide a balance between ASR and Visual changes. We have added a study on how the bounding norm affects the ASR. Since it respects linear growth it can be scaled up and down based on intended use.
>
>
> Thank you so much for taking the time to review our paper and we appreciate it a lot.
>
>
>
>
>
> [1]Dubey, S. R., & Singh, S. K. (2023). Transformer-based Generative Adversarial Networks in Computer Vision: A Comprehensive Survey. arXiv preprint arXiv:2302.08641.

---

### Author Response · Authors · 2023-05-30
**We opt-in for Archival**

We opt-in for Archival

---

### Comment · Area_Chair_4i14 · 2023-06-01
**Archival**

This work meets the threshold for archival, contents the URM statement and is deanonymized

---

### Meta-Review · Area_Chair_4i14 · 2023-04-02

**Recommendation:** Invite to archive
**Confidence:** 4

**Metareview:**

This paper has technical contributions on using a 3-encoder transformer-based GAN to attack vision transformers in a black-box setting. However, the reviewers are concerned with the originality of Transformer-based GANs, a lack of comparison with other GANs, a lack of the source of the improvement, and unclear details for reproducibility.


**Summary:**

This paper uses transformer-based GANs to attack vision transformers in a black-box setting.

**Reason For Not Giving A Higher Recommendation:**

The paper lacks comparison with existing works using transformer-based GANs, attacks with other GANs, and a further understanding on the improvement.



**Reason For Not Giving A Lower Recommendation:**

The paper has technical contributions on using transformer-based GANs to attack vision transformers.

---

### Decision · Program_Chairs · 2023-04-10

Invite to archive